# Poly(3-hydroxybutyrate-*co*-3-hydroxyvalerate) (P(3HB-*co*-3HV))/Bacterial Cellulose (BC) Biocomposites for Potential Use in Biomedical Applications

**DOI:** 10.3390/polym14245544

**Published:** 2022-12-18

**Authors:** Maria Râpă, Laura Mihaela Stefan, Ana-Maria Seciu-Grama, Alexandra Gaspar-Pintiliescu, Ecaterina Matei, Cătălin Zaharia, Paul Octavian Stănescu, Cristian Predescu

**Affiliations:** 1Faculty of Materials Science and Engineering, University Politehnica of Bucharest, 313 Splaiul Independentei, 060042 Bucharest, Romania; 2National Institute of Research and Development for Biological Sciences, 296 Splaiul Independentei, 060031 Bucharest, Romania; 3Advanced Polymer Materials Group, Faculty of Chemical Engineering and Biotechnology, University Politehnica of Bucharest, 1-7 Gh. Polizu Street, 011061 Bucharest, Romania

**Keywords:** poly(hydroxybutyrate-*co*-valerate), bacterial cellulose, melt processing, biocomposite, crystallinity, in vitro cytocompatibility

## Abstract

The aim of this study was to obtain biocomposites consisting of poly(3-hydroxybutyrate-*co*-3-hydroxyvalerate) (PHBV), bacterial cellulose (BC) and α-tocopherol by a melt processing technique for potential use in biomedical applications. The melt processing and roughness of biocomposites were evaluated and compared to sample without BC. The degradation rate of PHBV/BC biocomposites was measured in phosphate buffer saline (PBS) by determining the mass variation and evidencing of thermal and structural changes by differential scanning calorimetry (DSC) and attenuated total reflectance-Fourier transformed infrared spectrometry (ATR-FTIR). The cell viability, cell morphology, cell cycle distribution and total collagen content were investigated on murine NCTC fibroblasts. Overall, the adding of BC to polyester matrix led to an adequate melt processing of biocomposites and increased surface roughness and cytocompatibility, allowing the cells to secrete the extracellular matrix (collagen) and stimulate cell proliferation. Results showed that the PHBV/BC biocomposites were favorable for long-term degradation and could be used for the design of medical devices with controlled degradability.

## 1. Introduction

Polyhydroxyalkanoates (PHAs) are an important class of intracellular biopolyesters synthesized from more than 300 species of Gram-positive and Gram-negative bacteria, as an environmentally friendly alternative to petroleum-derived polymers [1]. The best-known representatives of the PHA class are poly(3-hydroxybutyrate) (P3HB) and poly(3-hydroxybutyrate-*co*-3-hydroxyvalerate) (P(3HB-*co*-3HV)), which have received an increased interest among researchers and manufacturers. These natural biopolyesters have been used to develop various medical devices used in tissue engineering [2,3], controlled drug delivery [4,5], orthopedics [6,7,8], wound healing [9,10] and vascular applications [2,11], as well as food packaging materials [12]. P(3HB) shows remarkable biodegradable and biocompatible properties, better resistance to UV degradation than polypropylene, water resistance, low toxicity due to its in vivo degradation to 3-hydroxybutyric acid (a metabolite found in human blood [13]) and easy processing, similar to conventional thermoplastic polymers. P(3HB) has a melting point (T_m_) in the range of 174–180 °C, a high degree of crystallinity (60 to 70%), and a glass transition temperature (T_g_) of 5 °C [14]. This combination of T_g_ and high crystallinity leads to obtaining very stiff and brittle formulations [15], limiting the practical medical applications of P(3HB). Therefore, the coatings and mixtures of P(3HB) with other polymers and additives represent a good strategy for manufacturing resorbable medical devices with lower production costs [16,17,18].

Poly(3-hydroxybutyrate-*co*-3-hydroxyvalerate) (P(3HB-*co*-3HV)), commonly known as PHBV, is also used for enhancing the P(3HB) properties, because it shows a lower melting temperature and crystallinity than P(3HB), and it is easier to process in the mold. In 1970, Imperial Chemical Industries Ltd. (Billingham, London, UK) marketed the P(3HB-*co*-3HV) product under the name Biopol^TM^ as a biodegradable thermoplastic material produced using the Gram-negative bacillus *Alcaligenes eutrophus*. However, the P(3HB-*co*-3HV) biopolymer displays a hydrophobicity, low degradation rate, inflammatory reaction and low cell proliferation as well as high-cost production [19,20], making it inadequate for short-term medical purposes. Cytocompatibility is an important feature to design certain medical devices, being influenced by the surface and interface properties of biocomposites [21]. The properties of PHA materials for medical formulations can be adjusted by varying the hydroxyvalerate (HV) content [22] and introduction of green fillers during processing [23,24]. The maximum degradation occurred for the P(3HB-*co*-3HV) matrix with a high content of HV.

Literature data shows various biomedical applications for PHA natural polyesters and their composites. PHBV was successfully mixed using the melt intercalation method with modified mineral clay (LDH-SDS) to create bionanocomposites with potential biomedical applications as dressings, demonstrated by in vitro biocompatibility tests (MTT and lactate dehydrogenase assays) conducted on human adipose-derived stem cells (hASCs) [25]. Modern wound dressings were proposed by incorporation of natural antioxidant agents such as olive leaf extract (OLE) in PHB/poly(hydroxyoctanoate-co-hydroxydecanoate) (PHB/PHOHD) nanofibers [26]. PHB/BC scaffolds prepared by melt mixing and leaching methods were found to be adequate for inducing new bone formation in adult CD1 mice [7]. Introduction of hydroxyapatite (HA) to PHB demonstrated excellent potential application as scaffolds for both hard (bone) and soft (nerve and cardiovascular) tissue regeneration [6]. By the coaxial electrospinning technique, the testing of PHBV/PCL-pullulan (core)/diatom scaffold (shell) highlighted the improved human osteosarcoma (Saos-2) cell viability, recommending its use for bone-tissue engineering [27]. In addition, by the incorporation of Ag nanoparticles or CuO nanoparticles in PHBV film, another possible medical application was found: virucidal activity against norovirus surrogates [28,29].

Bacterial cellulose (BC) is a sustainable versatile polysaccharide produced by at least 30 cellulose-producing bacteria [30]. It is a highly flexible biomaterial used to treat chronic wounds and burns, as artificial skin or wound dressings [31], carrier-scaffolds in cell-transference therapies [32] or for the preparing of active packaging [33]. Recently, commercial products based on BC have been available for wound dressing applications: BioFill^®^ or Dermafill™, Bionext^®^, Prima Cel™, Bioprocess^®^, Xcell^®^ etc. [34,35,36]. For this medical field, the purification, sterilization and non-toxic/non-infectious behavior of BC membranes are of utmost importance. The toxicology study of BC shows non-toxicity, as well as no inflammatory or oxidative stress responses at the cellular level provoked by ingestion, skin contact or inhalation of BC [37]. No foreign body reaction was observed by subcutaneous implantation of BC membranes and tubular shaped BC membranes in mice [34], and the absence of toxicity of BC nanofibers in vitro and in vivo tests are the proof of the safe use of this biopolymer in the medical field [38].

Vitamin E (±α-tocopherol) has been used as antioxidant agent to prepare bioactive food packaging formulations [39,40]. As well as assuring the active properties, the plasticizer effect of tocopherol for improving the thermal stability of PLA-based films [40] and ensuring the protection of ethyl cellulose films against environmental and biological stresses were reported [41]. Due to these features, vitamin E is recommended to be used in food packaging and wound care and drug delivery applications by melt processing technology [42,43].

The aim of this paper was to obtain bacterial-derived biocomposites by melt mixing of BC in the P(3HB-*co*-3HV) matrix in the presence of α-tocopherol, on machines similar to those used for conventional polymer processing and investigate their thermal processing, in vitro biodegradation and cytocompatibility. We explore the surface roughness, crystallinity degradation rate in PBS and cytocompatibility of PHBV biocomposites to find a potential use in tissue engineering applications.

## 2. Materials and Methods

### 2.1. Materials

The following raw materials were used to obtain the PHBV/BC biocomposites:(1)Polyhydroxybutyrate-co-valerate (P(3HB-*co*-3HV)) biopolymer, containing 12% hydroxyvalerate (HV) was supplied by Goodfellow Cambridge Ltd., Huntingdon, UK. The polymer matrix is obtained by biological fermentation from renewable carbohydrate feedstocks and is characterized by a density of 1.25 g/cm^3^, elongation at break of 35% and tensile strength at break of 23 MPa [44].(2)Bacterial cellulose (BC) was synthesized by the National Institute for Chemical–Pharmaceutical R&D (ICCF) Bucharest, Romania, in the form of fine powder, according to the procedure described by Codreanu et al. [7]. BC was obtained from renewable resources and it is characterized by remarkable mechanical properties such as porosity, water absorption, biodegradability and excellent biological affinity [45].(3)Vitamin E (±α-Tocopherol) was purchased from Sigma–Aldrich (St. Louis, MO, USA) and used as a plasticizer agent; it has a density of 0.95 g/cm^3^.

### 2.2. Methods

#### 2.2.1. Preparation of Bacterial-Derived Polymer Biocomposites

Two polymeric formulations based on P(3HB-*co*-3HB), 1 wt% vitamin E and BC in concentrations of 1 wt% and 2 wt%, respectively, were processed by melt mixing on the Brabender Plastograph provided with a 50 cm^3^ cuvette. The following working conditions were recorded: working temperature of 180 °C, mixing rate of 40 rpm and mixing time of 6 min. Before use, both P(3HB-*co*-3HV) granules and BC powder were dried in an oven (Memmert, Büchenbach, Germany) at a maximum temperature of 60 °C ± 1 °C for about 24 h. The dried materials were placed in a desiccator until use. The prepared biocomposites were denoted as PHBV/BC1% and PHBV/BC2%, respectively, depending on the bacterial cellulose content. In addition, the PHBV/Vitamin E sample containing P(3HB-*co*-3HV) (99%) and vitamin E (1 wt%) and a control (PHBV) were prepared in the same conditions as PHBV/BC biocomposites. The AS 220/X analytical balance with a density kit were used for the dosing of raw materials.

The physicochemical and biological properties were performed on thin films with the size of 200 mm × 200 mm × 0.1 mm, resulting from the pressing of polymeric formulation at a laboratory press (Polystat T200, COLLIN Lab & Pilot Solutions GmbH, Maitenbeth, Germany). The pressing conditions were: (i) preheating, for 5 min at a temperature of 180 °C; (ii) pressing, for 10 min at a temperature of 180 °C and a pressure of 147 bar and (iii) cooling, for 20 min at a pressure of 147 bar.

#### 2.2.2. Melt Processing Characteristics

The processing behavior of PHBV samples was investigated by analysis of torque, melt viscosity and energy consumption of the prepared samples. These processing parameters were calculated from torque–time curves recorded during melt blending in a Brabender Plastograph, at a working temperature of 180 °C, mixing rate of 40 rpm, and mixing time of 6 min.

#### 2.2.3. Atomic Force Microscopy

A 4000SPM MultiView/NSOM atomic force microscope system (AFM) (Nanonics Imaging LTD, Jerusalem, Israel) was used in non-contact mode to evaluate the surface morphology of the PHBV samples. Data acquisition was carried out by scanning probe microscopy (SPM) images collected from a sampling surface area of 80 µm × 80 µm or 40 µm × 40 µm, speed of 12 ms/point, and at a specimen distance ranging from 0.1 to 10 nm. Evaluation of the mean roughness of PHBV/Vitamin E and PHBV/BC biocomposites was achieved with the WSxM (4.0 software version number, Horcas, I; Fernandez, R., Gomez-Rodrigues, J.M.; Colchero, J.; Baro, A.M., Madrid, Spain).

#### 2.2.4. In Vitro Degradation

The evaluation of the degradability of the PHBV/Vitamin E, PHBV/BC1% and PHBV/BC2% biocomposites was performed by immersing specimens of 10 cm × 3 cm × 0.01 cm (cut from films) in fresh PBS (pH = 7.4), at a temperature of 37 °C, for different time intervals. The degradation of the materials was evaluated by determining the mass variation (analytical balance, precision of 0.1 mg) and investigation of thermal and structural changes by differential scanning calorimetry (DSC) and attenuated total reflectance-Fourier-transformed infrared spectrometry (ATR-FTIR). The specimens were tested after drying at 40 °C in an oven for 24 h and brought to constant mass before each determination of mass variation.

Mass loss was calculated using the following equation:(1)$$\mathrm{Mass}\;\mathrm{loss}\;(\%)=\frac{\left(M_{i}-M_{f}\right)\times 100}{M_{i}}\;$$
where *M_i_* and *M_f_* represent the initial mass and the final mass of the PHBV/Vitamin E and PHBV/BC specimens at time t in the PBS medium. The reported mass results are averages of a minimum of three specimen measurements ± standard deviation (SD).

Differential scanning calorimetry (DSC) thermograms of the materials before and after immersion in PBS were recorded on an 823^e^ DSC device (Mettler Toledo, Greifensee, Switzerland). The temperature profile was 30–200 °C and the heating rate was 5 °C/min. The calibration of DSC apparatus was performed using indium as a calibration standard (T_m_ = 156.6 °C; ∆H_m_ = 28.45 J/g). From the DSC, thermal diagrams were evaluated: melting temperature (Tm), melting enthalpy (ΔH_m_) and degree of crystallinity (X_c_), using the following equation:X_c_% = 100 × ΔH_m_/ΔH_m_^0^(2)
where ΔH_m_ is the enthalpy of mixing the samples (J/g); ΔH_m_^0^ is the heat of fusion for 100% crystalline PHBV (ΔH_m_^0^ = 146.6 J/g [19]). The mass fraction of neat PHBV in the composition of the mixtures was taken into account.

The spectral characteristics of the biocomposite films were recorded using an INTERSPEC 200-X Fourier transform infrared spectrophotometer (Interspectrum, Tartumaa, Estonia) with an ATR device, with an angle of incidence of 45°, in transmittance mode, the spectral domain of 4000–750 cm^−1^, and a resolution of 4 cm^−1^. The hydrogen-bonded C=O groups from samples were evaluated from the deconvolution of the FT-IR spectra between 1600 and 1800 cm^−1^. Neat PHBV was characterized by DSC and ATR-FTIR methods and served as a comparison for PHBV biocomposites before immersing in PBS.

#### 2.2.5. In Vitro Cytocompatibility Assessment

In vitro cytocompatibility tests were conducted on mouse fibroblast NCTC, clone 929 cell line according to the SR EN ISO 10993-5:2009 standard using the direct contact method. We evaluated the cell viability and morphology using the quantitative MTT assay and qualitative hematoxilin-eozin staining. The MTT assay is a colorimetric reaction measuring cell metabolic activity. This assay is based on the ability of mitochondrial succinate dehydrogenases of viable cells to reduce the tetrazolium dye 3-[4,5-dimethylthiazole-2-yl]-2,5-diphenyltetrazolium bromide (MTT) to purple insoluble formazan crystals. Briefly, cells were seeded in 24-well culture plates in minimum essential medium (MEM) supplemented with 10% fetal bovine serum (FBS) and 1% antibiotics (penicillin, streptomycin and neomycin) at a cell density of 4 × 10^4^ cells/mL, and maintained at 37 °C in a humidified atmosphere with 5% CO_2_ for 24 h to allow cell adhesion. The biocomposites were cut into discs of 5 mm in diameter and sterilized under UV light for 4 h. Samples were added into the wells (1 disk/well) and cells were further incubated in standard conditions for 24 and 48 h, respectively. After this period, the culture medium was replaced with 0.25 mg/mL MTT solution and plates were incubated for 3 h at 37 °C. Then, the insoluble formazan crystals formed were dissolved with isopropanol by gently stirring for 15 min at room temperature and the absorbance was measured at 570 nm using a Mithras LB 940 microplate reader (Berthold Technologies, Bad Wildbad, Germany). Untreated cells were used as negative control, whereas 0.003% hydrogen peroxide served as positive control. The concentration of converted dye directly correlates to the number of metabolically active cells. The results were calculated using the following equation:% cell viability = sample absorbance/negative control absorbance × 100

The negative control was considered 100% viable. All samples were tested in triplicate and data were calculated as mean values ± SD (*n* = 3).

For cell morphology analysis, mouse NCTC fibroblasts were seeded in 24-well culture plates at a cell density of 4 × 10^4^ cells/mL. After 24 h, samples were added into the wells (1 disc of 5 mm in diameter/well) and plates were incubated in standard conditions for 48 h. Cell morphology was evaluated using hematoxylin-eosin staining after cell fixation with Bouin’s solution. Light micrographs of fibroblast cells were obtained using a Zeiss Axio Observer D1 microscope and analyzed with AxioVision 4.6 software (Carl Zeiss, Jena, Germany).

#### 2.2.6. Cell Cycle Analysis by Flow Cytometry

For the analysis of the cell cycle, mouse NCTC fibroblasts were seeded at a density of 1 × 10^5^ cells/mL in 6-well culture plates in MEM supplemented with 10% FBS and incubated in standard conditions. After 24 h, MEM was replaced with fresh medium and samples were added into the wells (4 discs of 5 mm in diameter/well). After 24 h, cells were trypsinized, fixed in 70% ethanol overnight, washed with PBS and then incubated with 100 µg/mL RNase A (Promega, Madison, WI, USA) for 30 min at 37 °C followed by 50 µg/mL propidium iodide (PI—Becton Dickinson, Franklin Lakes, NJ, USA) for 30 min at 4 °C. Cell distribution in each phase of the cell cycle (G0/G1, S and G2/M) was expressed as a percentage and analyzed with BD LSR II flow cytometer and ModFit™ LT 3.0 software (Becton Dickinson, Franklin Lakes, NJ, USA).

#### 2.2.7. Total Collagen Assay

Total soluble collagen content was measured by Sircol collagen dye-binding assay [46] in order to evaluate the effect of the studied materials on the synthesis of extracellular matrix components. Briefly, mouse NCTC fibroblasts (5 × 10^4^ cells/mL) were incubated with the composite materials (4 discs of 5 mm in diameter/well), in 6-well plates at 37 °C for 5 days. 100 µL of cell supernatant was mixed with 1 mL Sircol reagent (0.1% Sirius Red in saturated picric acid solution) for 45 min with gentle stirring. The collagen-dye complex was centrifuged for 10 min at 10,000× *g* and the resulting pellet was dissolved in 250 µL alkali reagent (1 N NaOH). The absorbance was measured at 540 nm on a microplate reader Tecan Sunrise (Tecan, Grödig, Austria). Collagen quantification was determined using a standard curve for bovine acid-soluble type I collagen (Biocolor Assays, Carrickfergus, UK) with concentration ranging between 5 and 100 μg/mL.

## 3. Results

### 3.1. Melt Processing Evaluation

The incorporation of vitamin E into PHBV polymeric matrix led to enhancing the melt processability expressed as melt viscosity and energy consumption of biopolymer. Further, the addition of BC to PHBV copolymer led to a slow increase of the torque from 11 Nm, in the case of PHBV/Vitamin E, to 12 Nm and 13 Nm, for PHBV/BC1% and PHBV/BC2%, respectively (Table 1). This means a slow increase in the melt viscosity and energy consumption during the processing of PHBV/BC biocomposites, but not higher than the processing characteristics of neat PHBV. Without the contribution of vitamin E to improve the flexibility of PHBV chains, the processing of PHBV/BC biocomposites would have been difficult. Similar values for melt viscosities were reported in the case of PHB plasticized with 20% tributyl citrate (TBC) and filled with max. 2 wt% BC [47].

### 3.2. Atomic Force Microscopy (AFM)

Two-dimensional and three-dimensional topographic images for PHBV/BC biocomposite films recorded by atomic force microscopy (AFM) are depicted in Figure 1, whereas root mean square roughness (RMS), defined as the standard deviation of height above an average plane, is shown in Table 2.

The surface roughness of PHBV/BC biocomposites increased compared to that of the PHBV/Vitamin E sample, increasing with BC content, indicating a poor dispersion of BC into the PHBV matrix. This is due to the weak interface between the PHBV matrix and filler. AFM analysis for PHBV/BC2% biocomposite exhibited a “sea-island” morphology. The vitamin E contributes to a relatively smooth surface of PHBV film, assuring a better dispersion in the polymeric matrix. These results are correlated with the processing parameters reported in Table 1.

### 3.3. In Vitro Degradation

#### 3.3.1. Mass Loss

The mass loss of the mixtures based on PHBV and BC, immersed in PBS, at different time intervals is shown in Figure 2, and it can be observed that it was influenced by the sample composition. The degradation of PHBV/BC biocomposites started immediately after the immersion of specimens in the PBS medium and it was attributed to the solvent movement in the spaces between the macromolecular chains, favoring the hydrolysis of the surfaces and, later, the degradation. The sample containing 2% BC recorded the highest mass loss (1.12 ± 0.04%), after 30 days of immersion.

#### 3.3.2. Thermal Analysis by DSC

Figure 3 revealed the DSC curves for biocomposites based on PHBV, BC and vitamin E, both initially and after immersion in PBS for 30 days. The melting temperature (T_m_), enthalpy of fusion (ΔH_m_) and degree of crystallinity (X_c_) were evaluated from the DSC curves, before and after exposure in PBS. The values of these thermal parameters are mentioned in Table 3.

Figure 3 shows the appearance of two endothermic melting peaks for the tested biocomposites samples associated with the melting of different types of crystallites from the PHBV structure [48,49]. The first and smaller endothermic peak of PHBV sample occurred at T_m_ of 149.2 °C and is assigned to the melting of low stability crystals, while the another melting peak located at 162.5 °C is due to the melting of the large and more perfect crystals [48]. Data from Table 3 shows that both melting temperature peaks decreased for PHBV/BC1% and PHBC/BC2% biocomposites as compared with PHBV sample, suggesting the generation of immiscible biocomposites. In addition, the introduction of BC affects the degree of crystallinity (X_c_) of PHBV, which decreased for PHBV/BC1% and PHBV/BC2% biocomposites compared to the control sample, due to the increased interfacial area. The reducing trend for melting enthalpy and crystallinity was also reported by Zheng et al. [50] in the case of PHBV/cellulose nanocrystals (CNCs) composites. The PHBV/BC2% biocomposite showed an increase in the degree of crystallinity compared to PHBV/Vitamin E and PHBV/BC1% biocomposites, associated with the possible chain irregularity of BC and agglomeration during melt processing. The degree of crystallinity of PHBV/Vitamin E sample decreases with respect to that of neat PHBV. This decrease is attributable to the vitamin E, which improves the flexibility of macromolecular chains but reduces the mobility of crystallizable segments from the PHBV chain. This behavior is in good agreement with the lower melt viscosity reported in Table 1. The degree of crystallinity for PHBV/Vitamin E, PHBV/BC1%, and PHBV/BC 2% biocomposites increased after immersion in PBS compared to unexposed samples, by ~61%, 63% and 27%, respectively, in accordance with the mass loss in time presented in Figure 2.

#### 3.3.3. Fourier-Transform Infrared Spectroscopy-Attenuated Total Reflectance (FT-IR–ATR)

The spectral characteristics of the PHBV/BC biocomposites were analyzed both initially and after immersion in PBS by Fourier-transform infrared spectroscopy with ATR technique.

The FT-IR spectra recorded for the PHBV and BC based samples showed absorption bands of these components at the following wavenumbers: 1720 cm^−1^ (hydrogen-bonded of the C=O groups from the crystalline group of PHBV), 1276 cm^−1^ and 1054 cm^−1^ (C-O-C stretching vibration), 1180 cm^−1^ (C-O-C stretching vibration in the amorphous region) and 1456 cm^−1^ (CH_2_ bending vibration) (Figure 4a), according to the literature data [51,52]. A small intensity of –CH stretching and C-H stretching vibration are observed at 2983 cm^−1^ and 2931 cm^−1^. Small changes in the intensity and the wider profile of the absorption bands are observed at around 1720 cm^−1^, 1269 cm^−1^, (–C-O-C stretching mode of crystalline part) and 1182 cm^−1^ (amorphous state of C-O-C stretching band). These indicate that intermolecular hydrogen-bonding interactions between O−H groups of BC and carbonyl groups of poly(3HB-*co*-3HV) matrix occurred.

Compared to the FT-IR spectrum recorded for the control sample (PHBV), it was found that the intensities of the three crystalline bands from 970 cm^−1^, 898 cm^−1^ and 828 cm^−1^ (β-linked glucose-based polymers) decreased for unexposed PHBV/BC1% and PHBV/BC2% biocomposites, as well as for those exposed to the PBS action (Figure 4b). Furthermore, the area of the absorption band of C=O groups increased for PHBV/BC biocomposites compared to the PHVB/Vitamin E (Figure 4c). The immersion of samples into PBS medium led to the displacement of the carbonyl stretching band to around 1723 cm^−1^ and the increased area for PHBV control and PHBV/BC1% as compared to unexposed samples, indicating that the chemical degradation occurred. These results were in good correlation with the degree of crystallinity shown in Table 3. A remarkable decrease in the crystallinity band area was observed when comparing the PHBV biocomposites to PHBV/Vitamin E (Figure 4d), suggesting that the mixture of BC provoked the resistance of the amorphous domains of PHBV to the hydrolytic degradation.

#### 3.3.4. In Vitro Cytocompatibility Assessment

The results of the MTT assay showed a high degree of cytocompatibility of all tested biocomposites at both exposure times (Figure 5). After 24 h of treatment, the values of cell viability ranged between 86.67% for PHBV/BC1% and 96.33% for PHBV/Vitamin E. After 48 h, an increase of cell viability was observed for all samples, with values close or even higher than that of the negative control. The highest value of cell viability was recorded for PHBV/Vitamin E (104.77%), whereas the lowest for PHBV/BC2% biocomposite (98.81%).

The morphological aspect of the cells cultivated in the presence of polymeric biocomposites was investigated under an optic microscope after hematoxylin-eosin staining. The negative control exhibited a normal phenotype, with cells having a polygonal shape, euchromatic nuclei, clear cytoplasm and cytoplasmic extensions (Figure 6e). The morphological appearance of cells treated with PHBV and PHBV/BC biocomposites was similar to that of the negative control (Figure 6a–d). In addition, for all samples, cell density was comparable to that of the negative control, with cells covering around 80–90% of the culture surface. Clear morphological changes in the cellular shape, cell membrane, cytoplasm and nuclei were observed in the case of the positive control, where cell viability was around 10% (Figure 6f).

#### 3.3.5. Cell Cycle Distribution

Cell cycle distribution was investigated by flow cytometry in order to evaluate cell proliferation after the treatment with different variants of PHBV/BC-based biocomposites for 24 h. This method is based on the quantification of the DNA content of the treated cells, indicating potential changes in the cell population distribution between the three phases of the cell cycle. In our study, the obtained histograms showed a cell cycle distribution similar to that of the control (untreated cells) for all tested samples, indicating a normal cell growth rate and no negative effect on cell proliferation (Figure 7).

The tested biocomposites induced a slight decrease of the cell population in the G0/G1 phase compared to the control (73.35%), ranging between 70.65% for PHBV/Vitamin E and 71.89% for PHBV/BC2% (Table 4). Furthermore, the percentage of cells found in the S phase increased compared to the control (13.57%), ranging between 15.51% for PHBV/Vitamin E and 15.92% for PHBV/BC1%. The same pattern was also observed for the cell population in the G2/M phase, with higher percentages found for tested samples (ranging between 13.62% for PHBV/BC2% and 14.21% for PHBV/Vitamin E) when compared to the control (12.87%) (Table 4).

#### 3.3.6. Total Collagen Content

In this study, Sircol assay was performed in order to estimate the quantity of the newly synthesized collagen by mouse NCTC fibroblasts in the presence of the composite materials during 5 days of cultivation. The obtained results showed that all samples were able to stimulate collagen production by fibroblasts to a greater extent, with values ranging from 0.401 ± 0.03 mg/mL for PHBV/BC2% to 0.479 ± 0.02 mg/mL for PHBV/Vitamin E, compared to the control (0.346 ± 0.02 mg/mL) (Table 5).

## 4. Discussion

The viscosity of the melted polymers and their dependence on temperature and shear rate are significant factors in terms of processing technologies at large scale which involve melt flow, such as pressing, extrusion and injection-molding. It is known that the extruder cylinder pressure is dependent on the melt viscosity of the molten polymer. Most often, a low melt viscosity is an indication of the ease of melt flow during the extrusion process. Usually, the plasticizers are added during melt processing of biopolymers with the purpose of decreasing the viscosity of samples. In this study, the adequate melt processing of biocomposites was achieved by using a poly(3HB-*co*-3HV) copolymer and vitamin E. The poly(3HB-*co*-3HV) copolymer has a good melt processability as compared to that of PHB. This can be explained by the structure of the copolymer, which contains 12 mol% HV, acting as a plasticizer [53]. The adding of vitamin E enhanced the processing characteristics of the PHBV natural polyester, while the addition of BC to the polymer matrix did not lead to hard processing conditions.

The topography of PHBV/BC biocomposites was evaluated by an AFM technique. The roughness’ surface plays an important role for cell viability and cell proliferation as well as in vitro degradation, being an important parameter for the designing of biomaterials for medical applications. In this study, the roughness of PHBV/vitamin E was 13.9 nm. The incorporation of a BC filler led to an increased roughness surface of the PHBV/BC biocomposite, but maintained in nanometer scale. In a comparative study about the roughness surface of PHBV and PHB films fabricated by a solvent casting method performed by Mohandas et al. [53] an average surface roughness of 43.66 nm and 67.93 nm, respectively, was demonstrated. Other authors also reported the highest value of surface roughness for PHB/BC scaffolds (129 nm) [47], PLA/collagen/AgNPs bionanocomposite films (57 nm) processed by melt technique [54,55] and PLA reinforced with thermoplastic cassava starch (TPCS) and commercial graphene (GRH) nanoplatelets (141.8 nm) processed by reactive blending [56]. The increased roughness could be attributed to the irregular shape of filler. It is well known that an uniform dispersion of fillers in a matrix is a first condition for the formation of more intermolecular hydrogen bonds in the nanocomposites [19] with a consequence of a decrease in the degree of crystallinity and the improvement of cell proliferation.

The experimental design on the degradation of a biomaterial should include a study about the mass loss, deterioration of mechanical properties, increase in the degree of crystallinity and loss of structural integrity [57]. During P(3-HB-*co*-HV) copolymer degradation in PBS medium for 12, 14 and 44 weeks it was observed that the molecular mass decreased to 212 kDa, 194 kDa and 121 kDa, respectively [58]. The advantages of using PHBV for medical applications rather than biomaterials from PLA are related to a slower and controlled degradation of PHBV biomaterials, being favorable for tissue engineering applications. The obtained results indicated that the PHBV/BC2% biocomposite was favorable for long-term degradation, and could be used for the design of medical devices with controlled degradability. In short-term use (1–2 days), the tested PHBV biocomposites exhibited a steady degradation. In vitro degradation of PHBV/BC biocomposites occurred with the degradation of the amorphous part of the polymers due to the effect of hydrolytic scission of polymer chains [58]. The low variation of the degree of crystallinity for PHBV/BC2% biocomposite after immersion to PBS medium can be explained by the low dispersion in polymeric matrix during melt processing. The addition of BC improved the mass loss recorded for PHBV/BC biocomposites, which is correlated with the surface roughness values of samples, attributed to the migration of BC from biocomposites. Similar behavior was reported for PHBV/highly crystalline cellulose nanoparticles (CNPs) composites, and explained by the creation of a barrier layer due to the rigid hydrogen-bond network between components, able to protect the functional groups of PHBV [19]. Other authors also reported the decreased crystallinity of PHBV composites containing hydroxyapatite nanoparticles after PBS immersion [58]. A slower in vitro degradation of PHBV/wollastonite composite scaffolds than that encountered for pure PHBV was observed by Li et al. [59], which was assigned to the acid degradation of PHBV caused by the dissolution of alkaline ions from the wollastonite. The bulk degradation is the predominant mechanism occurring for PHBV samples [57].

In this study, a biological characterization of the PHBV/BC biocomposites using different approaches, such as quantitative MTT assay, qualitative cell morphology evaluation, cell cycle distribution and quantification of the total collagen synthesized by the cells after the treatment with the tested samples was also performed. Preliminary in vitro biological assessment showed that PHBV/BC biocomposites were cytocompatible, induced the cells to secrete extracellular matrix (collagen) after 5 days of cultivation and also stimulated cell proliferation, since most of the cells were not blocked in any phase of the cell cycle. Our findings were in agreement with previous studies that have reported different scaffolds based on PHBV or BC with good biological properties, making them suitable for tissue engineering [60,61,62,63,64]. For example, a biomaterial based on bacterial nanocellulose, produced by the *Komagataeibacter xylinus*, provided structural support for fast and stable adherence of the mesenchymal stem cells and also stimulated their osteogenic differentiation [64]. Different BC composites blended with chitosan, gelatin, poly(ethylene glycol) (PEG) or natural poly(3-hydroxybutyrate) (PHB) also showed a good biocompatibility and promoted cell adhesion and proliferation, suggesting the use of these scaffolds for biomedical purposes, including wound dressing [62,65]. In order to improve the antimicrobial activity of BC for wound healing, new BC hybrid gel-membranes were developed with the addition of Ag nanoparticles. These hybrid nanostructures exhibited relatively low in vitro cytotoxicity because of the slow release of Ag ions, allowed the growth and attachment of epidermal rat cells and also supported cell differentiation into fibroblasts [63]. When looking at improved biomaterials based on PHBV as potential candidates for biomedical applications, in vitro tests conducted on bioactive ternary composites containing PHBV, nano magnesium calcium phosphate (n-MCP) and zein (ZN) showed that these scaffolds were cytocompatible, provided suitable niches for cell attachment and spread, promoted cell proliferation and enhanced alkaline phosphatase (ALP) activity of MC3T3-E1 cells, used as a marker for osteoblast differentiation [66]. Other functional scaffolds, such as PHBV microspheres embedded in PLGA matrix or electrospun composites containing PHBV, silk fibroin and nano-hydroxyapatite, were shown to support cell attachment and proliferation, and, therefore, are suitable for bone tissue engineering [61,67]. Other studies have reported the fabrication by electrospinning of nanofibrous scaffolds based on PHBV and type-I collagen or PHBV and keratin, and preliminary cell culture experiments indicated that NIH3T3 cells exhibited a higher capacity to adhere and proliferate onto these scaffolds compared to the PHBV nanofibrous scaffold [60,68]. Other commercial biopolymers, such as PLA (NatureWorks LLC, Plymouth, MN, USA) [54,55,69], PHB (Biomer, Krailling, Germany) [7] and PHBV (Good Fellow Cambridge Limited, Huntingdon, UK) [25] as well as their biocomposites, indicate in vitro biocompatibility, for further use in various biomedical applications [70].

Generally, high levels of collagen production are often associated with enhanced fibroblast proliferation [55,71], our results being consistent with the cytocompatibility values measured at 48 h. A previous study reported that bacterial nanocellulose stimulated collagen networks formation by mesenchymal stem cells [64]. However, in our study, increasing concentration of bacterial cellulose in the material composition induced a slight decrease of collagen synthesis by NCTC cells compared to the PHBV composite, suggesting that the content of BC has no important effect on collagen synthesis. Nevertheless, the obtained results showed that all tested composite materials stimulated cell proliferation and collagen production compared to the untreated cells. In addition, our results on cell cycle distribution were also in agreement with other studies, which reported similar distribution of cell populations between the three phases of the cell cycle for different biomaterials, such as PLA-collagen-AgNPs based bionanocomposites [54]. Cell cycle results correlated well with the rest of biological measurements, such as MTT assay, cell morphology evaluation or quantification of total collagen content, indicating a good cytocompatibility for all tested biocomposites and their potential use as biomedical devices for tissue engineering.

## 5. Conclusions

The studied formulations based on poly(3-hydroxybutyrate-*co*-3-hydroxyvalerate) (P(3HB-*co*-3HV)), bacterial cellulose (BC) and vitamin E were prepared by melt processing, using the same equipment as those used for processing of synthetic polymers. The incorporation of BC led to a slow increase in melt processing, together with the increased surface roughness of the PHBV biocomposites. These features induced the cells to secrete extracellular matrix (collagen) and to stimulate cell viability and proliferation. Vitamin E acted as a plasticizer for PHBV matrix. The results showed that PHBV/BC biocomposites could be favorable for long-term degradation, indicating the potential to be used for medical devices with controlled degradability. In short-term use (1–2 days) the tested biocomposites exhibited a steady degradation. Further tests are needed to assess in vivo biocompatibility as well as the bioactivity of PHBV/BC biocomposites for a certain medical field.

## Figures and Tables

**Figure 1 polymers-14-05544-f001:**
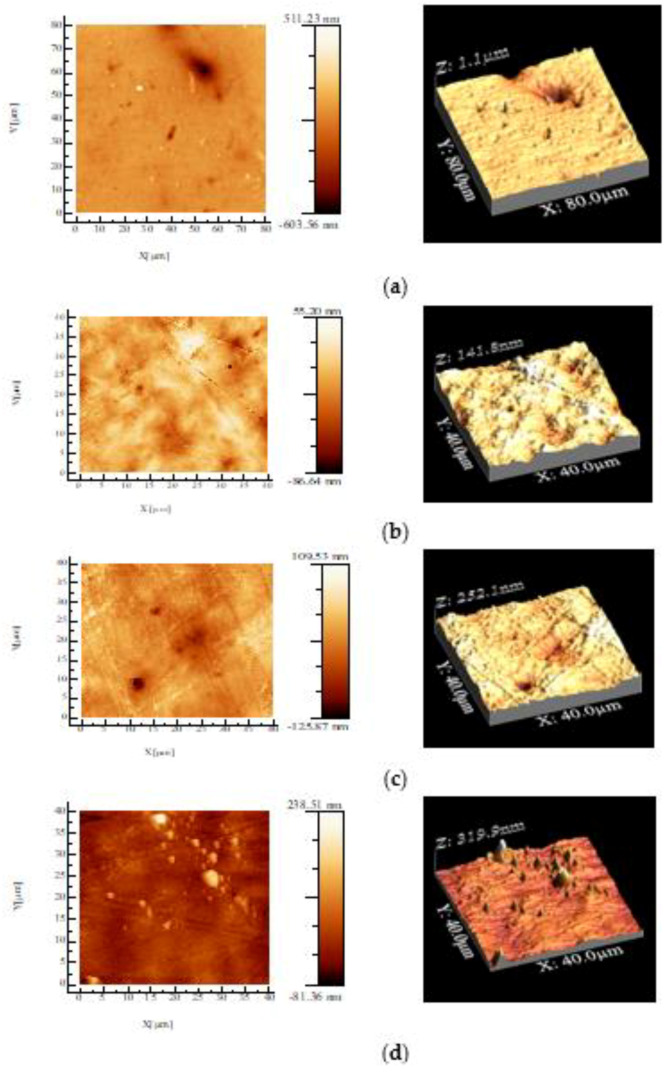
Atomic force microscopy images of PHBV/BC biocomposite films and PHBV film with scanning size of 40 µm × 40 µm and 80 µm × 80 µm, respectively. PHBV neat (**a**); PHBV/Vitamin E (**b**); PHBV/BC1% (**c**); PHBV/BC2% (**d**).

**Figure 2 polymers-14-05544-f002:**
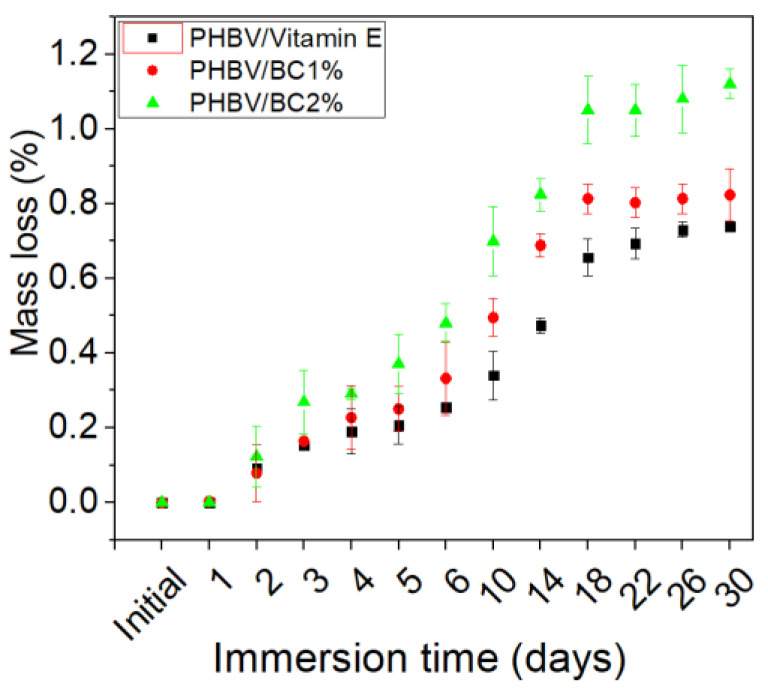
Mass loss of PHBV/BC biocomposites immersed in PBS as a function of exposure time (after 30 days in vitro hydrolytic degradation).

**Figure 3 polymers-14-05544-f003:**
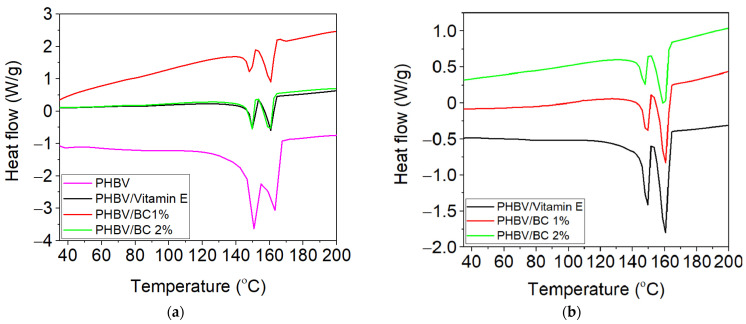
DSC curves for biocomposites based on PHBV, BC and vitamin E, first heating (**a**) initial; (**b**) after immersion for 30 days in PBS at 37 °C (exo up).

**Figure 4 polymers-14-05544-f004:**
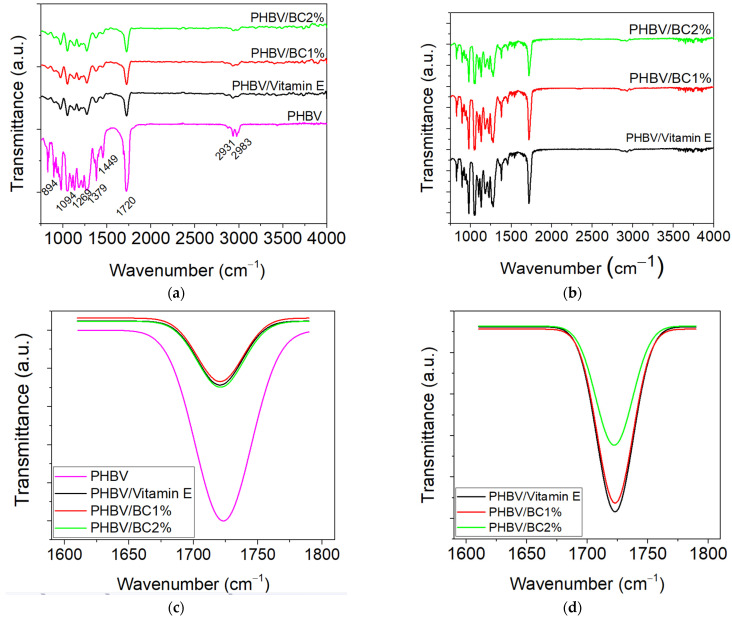
ATR-FTIR normalized spectra for biocomposites based on PHBV, BC and vitamin E. (**a**) initial; (**b**) after 30 days immersion in PBS; (**c**) carbonyl stretching region (–C=O) for initial samples; (**d**) carbonyl stretching region (–C=O) for degraded samples in PBS.

**Figure 5 polymers-14-05544-f005:**
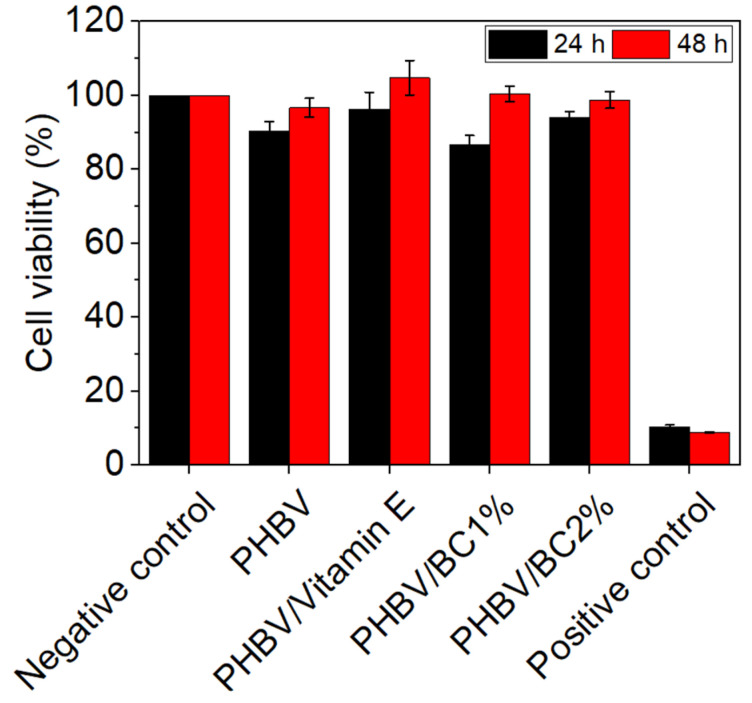
Viability of mouse NCTC fibroblasts cultivated in the presence of PHBV and BC biocomposites for 24 and 48 h, evaluated by the MTT assay. Samples were reported to the negative control (untreated cells) considered to have a 100% viability. Data were expressed as mean values ± SD (*n* = 3).

**Figure 6 polymers-14-05544-f006:**
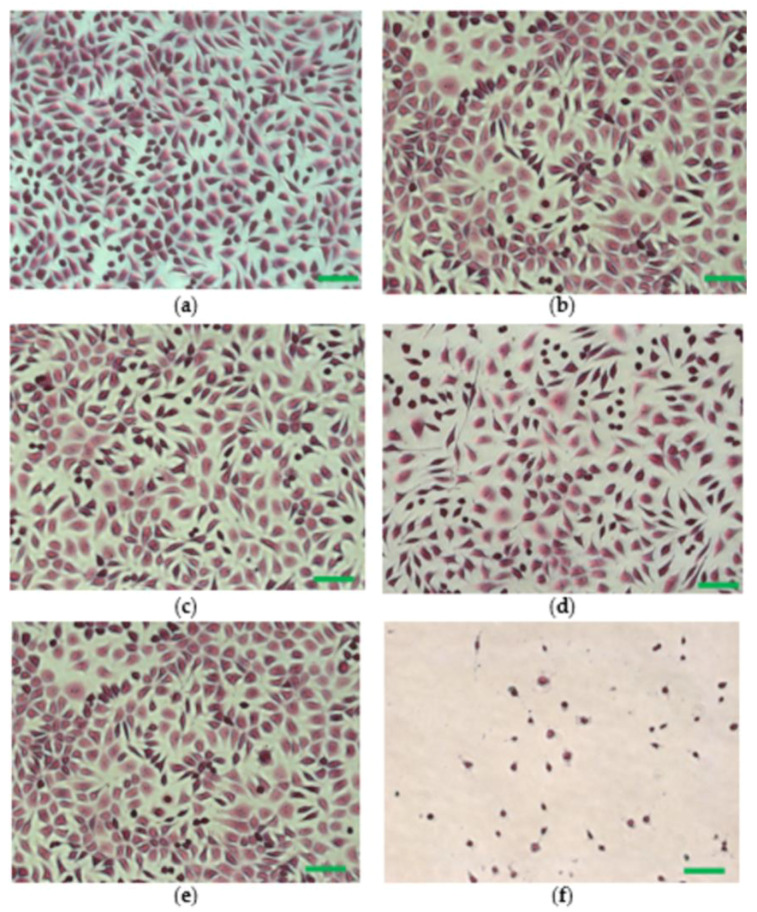
Cell morphology after 48 h treatment with different biocomposites: neat PHBV (**a**); PHBV/Vitamin E (**b**); PHBV/BC1% (**c**); PHBV/BC2% (**d**). Negative control was represented by untreated cells (**e**) and positive control by cells cultivated in MEM containing 0.003% H_2_O_2_ (**f**). Scale bar = 50 μm (hematoxylin-eosin staining).

**Figure 7 polymers-14-05544-f007:**
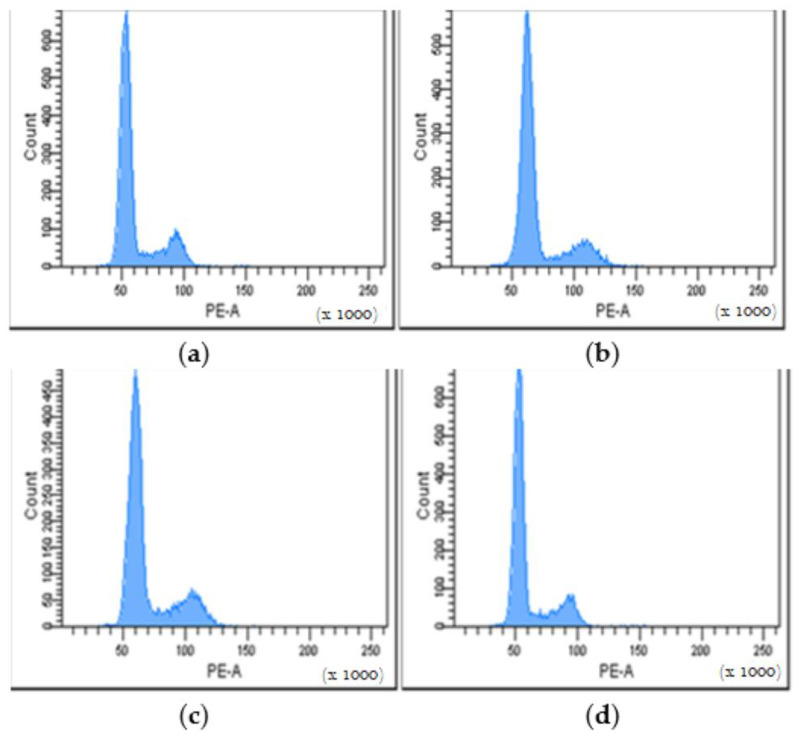
Cell cycle histograms of mouse NCTC fibroblasts treated with different PHBV/BC-based biocomposites for 24 h. Control (**a**); PHBV/Vitamin E (**b**); PHBV/BC1% (**c**); PHBV/BC2% (**d**).

**Table 1 polymers-14-05544-t001:** Melt processing parameters for PHBV/BC biocomposites compared with neat PHBV evaluated from torque-time curves.

Processing Parameter	PHBV	PHBV/Vitamin E	PHBV/BC1%	PHBV/BC2%
Torque at 1 min (Nm)	16	15	26	30
Torque at 2 min (Nm)	25	26	30	32
Torque at 3 min (Nm)	17	15	28	28
Torque at 4 min (Nm)	16	14	23	24
Torque at 5 min (Nm)	14	12	18	19
Torque at 6 min (Nm)	13	11	12	13
Melt viscosity (Nm/rpm)	0.325	0.275	0.300	0.325
Energy consumption (W)	54.42	46.05	50.24	54.42

**Table 2 polymers-14-05544-t002:** Roughness values for the PHBV/BC biocomposites compared with PHBV evaluated from AFM determination.

Sample	AFM Roughness Data (nm)
PHBV	15.5
PHBV/Vitamin E	13.9
PHBV/BC1%	22.0
PHBV/BC2%	25.7

**Table 3 polymers-14-05544-t003:** DSC parameters obtained for PHBV/BC biocomposites as compared with neat PHBV.

Sample	Initial			After 30 Days Immersion in PBS		
	ΔH_m_ (J/g)	T_m_ (°C)	X_c_ (%)	ΔH_m_ (J/g)	T_m_ (°C)	X_c_ (%)
PHBV	58.9	162.5 149.2	40.3			
PHBV/Vitamin E	43.4	160.8 150.2	30.0	70.1	160.8 148.9	48.5
PHBV/BC1%	34.8	160.4 148.7	24.3	56.9	160.0 148.8	39.8
PHBV/BC2%	45.7	160.0 149.3	32.2	57.9	159.7 147.3	40.9

**Table 4 polymers-14-05544-t004:** Cell cycle distribution (%) of mouse NCTC fibroblasts treated with PHBV/BC-based biocomposites.

Sample	G0/G1 (%)	S (%)	G2/M (%)
Control	73.35	13.57	12.87
PHBV/Vitamin E	70.65	15.51	14.21
PHBV/BC 1%	70.74	15.92	13.65
PHBV/BC 2%	71.89	15.75	13.62

**Table 5 polymers-14-05544-t005:** Total soluble collagen secreted by mouse NCTC fibroblasts in the presence of the biocomposite materials for 5 days.

Sample	Total Collagen (mg/mL)
Control	0.346 ± 0.02
PHBV/Vitamin E	0.479 ± 0.02
PHBV/BC1%	0.434 ± 0.04
PHBV/BC2%	0.401 ± 0.03

## Data Availability

Not applicable.

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
