# Peer review of "Poly(3-hydroxybutyrate-co-3-hydroxyvalerate) (P(3HB-co-3HV))/Bacterial Cellulose (BC) Biocomposites for Potential Use in Biomedical Applications"

_polymers, 2022, doi:10.3390/polym14245544_

Round 1

Reviewer 1 Report

The manuscript presented here is about the preparation of environmentally friendly biocomposites by melt processing method. PHBV and BC were used as the main components of this biocomposite material. The effect of BC was investigated and α-tocopherol was loaded in these biocomposites. Obtained materials were characterized by a variety of aspects, and their biocompatibility was evaluated against the L929 cell line, for which ECM production was demonstrated with respect to collagen amount. Its slow degradation rate was presented as an advantageous tool for its applications in medical device fabrications. Overall, the manuscript was written in detail, with adequate literature background. The following issues would be addressed in the manuscript for helping more clarity.

*Is there any data or reference showing the non-toxic / non-infectious behavior of synthesized and purified BC on the bulk scale?

*For Fig.2, is there a meaningful difference between the swelling behavior of only PHBV and BC-containing ones (like p-value)?

*Is there any control for the biocomposites without α-tocopherol, to understand its effect on cells‘ metabolic activities?

*How is the intensity of peaks in ATR*FTIR related to its amount? Which peak was taken as a reference or fixed?

Author Response

*Is there any data or reference showing the non-toxic / non-infectious behavior of synthesized and purified BC on the bulk scale?

The Introduction section was completed, as follows:

“Recently, commercial products based on BC have been available for wound dressing applications: BioFill® or Dermafill™, Bionext®, Prima Cel™, Bioprocess®, Xcell® [34-36]. For this medical field, the purification, sterilization and non-toxic / non-infectious behavior of BC membranes are of utmost importance. The toxicology study of BC shows non-toxicity, as well as no inflammatory or oxidative stress responses at the cellular level provoked by ingestion, skin contact or inhalation of BC [37]. No foreign body reaction was observed by subcutaneous implantation of BC membranes and tubular shaped BC membranes in mice [34], and the absence of toxicity of BC nanofibers in vitro and in vivo tests are proofs for safe use of this biopolymer in medical field [38]. “

*For Fig.2, is there a meaningful difference between the swelling behavior of only PHBV and BC-containing ones (like p-value)?

By performing the Anova Single Factor analysis at a significance level (p) of 0.05, we concluded that there is no statistically significant difference in mean of samples.

*Is there any control for the biocomposites without α-tocopherol, to understand its effect on cells‘ metabolic activities?

We included data on cell viability and cell morphology for neat PHBV in Result section, according to Figure 5 and Figure 6 (a). PHBV biopolymer shows a cell viability of 90.42% and 96.68% at 24 h and 48 h, respectively.

*How is the intensity of peaks in ATR*FTIR related to its amount? Which peak was taken as a reference or fixed?

The intensity of peaks in ATR-FTIR analysis decreased with the introduction of BC into PHBV matrix, as can be observed from Figure 4. Figure 4 contains the FTIR spectrum of neat PHBV.  The peak at 1456 cm-1 (CH2 bending vibration) was taken as a reference peak.

Reviewer 2 Report

The manuscript is sound and well- organized, it could be accepted after following improvements:

1. AFM roughness implication is not demonstrated, how it being affected and how it effecting on other performances for biomedical applications.

2. Cytocompatibility needs to include the principle, calculation and explanations. BTW, some matured commercial polymer sample could be compared herein.

3.Both in introduction and discussion sections, the author is suggested to expand and further the product's biomedical application concerning significance and how to be used. Current text is not adequate enough, and the logic connection between them is weak. please improve accordingly.

4.Some references on ht derivatives and similar polymers are suggeted to be added in.

Author Response

The manuscript is sound and well- organized, it could be accepted after following improvements:

  1. AFM roughness implication is not demonstrated, how it being affected and how it effecting on other performances for biomedical applications.

The AFM for neat PHBV was performed and introduced in Figure 1 and the roughness value was calculated in Table 2.

The discussion section was completed as follows:

“The roughness’ surface plays an important role for the cell viability and cell proliferation as well as in vitro degradation, being an important parameter for designing of biomaterials for medical applications. In this study, the roughness of PHBV/Vitamin E was 13.9 nm. The incorporation of BC filler led to an increased roughness surface of PHBV/BC biocomposited, but maintained in nanometer scale.”

“In our study, the roughness’ surface value increased with incorporation of BC, leading to the slowly decrease in the cell viability values, but maintaining in the acceptable limit of biocompatibility imposed by standards SR EN ISO 10993-5:2009.”

“The addition of BC improved the mass loss recorded for PHBV/BC biocomposites, which is correlated with the surface roughness values of samples, attributed to the migration of BC from biocomposites.”

  1. Cytocompatibility needs to include the principle, calculation and explanations. BTW, some matured commercial polymer sample could be compared herein.

We included in the section of Materials and Methods the principle of the MTT assay and the equation (3) for the calculation of the viability percentages.

Discussion section was completed as follows:

“Other commercial biopolymers, like PLA (NatureWorks LLC, Minnetonka, USA) [54,55,69] and PHB (Biomer, Krailling, Germany) [7], and PHBV (Good Fellow Cambridge Limited, Huntingdon UK) [25] as well as their biocomposites indicate in vitro biocompatibility, for further use in various biomedical applications [70].“

  1. Both in introduction and discussion sections, the author is suggested to expand and further the product's biomedical application concerning significance and how to be used. Current text is not adequate enough, and the logic connection between them is weak. please improve accordingly.

The Introduction text was revised, old references were removed and new applications of PHA and their composites were added.

“Literature data show various biomedical applications for PHA natural polyesters and their composites. PHBV was successfully mixed melt intercalation method with modified mineral clay (LDH-SDS) to create bionanocomposites with potential biomedical application for dressings demonstrated by in vitro biocompatibility conducted with human adipose derived stem cells (hASCs), MTT and lactate dehydrogenase assays [25]. Modern wound dressings were proposed by incorporation of natural antioxidant agents such as olive leaf extract (OLE) in PHB/poly(hydroxyoctanoate-co-hydroxydecanoate) (PHB/PHOHD) nanofibers [26]. PHB/BC scaffolds prepared by melt mixing and leaching methods were found as adequate for inducing new bone formation to adult CD1 mice [7]. Introduction of hydroxyapatite (HA) to PHB demonstrated excellent potential application of as scaffolds for both hard (bone) and soft (nerve and cardiovascular) tissue regeneration [6]. By coaxial electrospinning technique, the testing of PHBV/PCL-pullulan (core)/diatom scaffold (shell) highlight the improved human osteosarcoma (Saos-2) cell viability,  recommending  its use for bone-tissue engineering [27]. Also, by incorporation of Ag nanoparticles or CuO nanoparticles in PHBV film, another possible medical application was fond as virucidal activity against norovirus surrogates [28,29].”

  1. Some references on ht derivatives and similar polymers are suggeted to be added in.

We added new references about the other derivatives and similar polymers and their biocomposites according to point 3.

Reviewer 3 Report

Concerning the language, the manuscript is well written. Only, the title is not clear: what is the role of the word environmentally? Is something missing? Tables and figures (except the ordinate of figure 2) is well presented, too. Techniques applied are appropriate.  The idea is worth of investigation, unfortunately the results are not worth of publishing. Why? The targets pointed out in lines 98-104 were not achieved. Influence of tocopherol on compatibilization and plasticization, if there was no sample without tocopherol, is impossible to established. Other properties were worsened compared to control sample: processing properties, roughness, mass loss as well as the collagen production. Decrease of the glass transition temperature rather than the melting point is the measure of plasticization action. Let’s go step by step: 1.      All samples contain 1 % of tocopherol, so the influence of tocopherol is not possible to determine. The only parameter that changes is the BC content. 2.      From table 1 is obvious that addition of BC doesn’t improve the melt processing. 3.      Also, from the table 2 is obvious that BC addition increase the surface roughness. 4.      Table 2 shows that with increase in BC content weigh loss increase, too. 5.      DSC parameters (Table 3), except crystallinity, do not show significant changes. Taking in to account method accuracy and experimental errors, melting points are the same for all samples, as well as before and after immersion in PBS. Melting temperature decrease in not a measure of plasticization effect, the glass transition temperature is. But, as I wrote at the beginning, all of your samples contain tocopherol in the same amount. The melting point depression is the result of BC addition and it is normal, because BC can influence the copolymer crystallization. 6.      All samples have almost the same cell compatibility, but the values of secreted collagen decrease with BC content. 7.      Discussion in the lines 421-422 is not the result of presented investigations: melt viscosity was increased; energy consumption was higher… 8.      23 % of cited references are older than ten years and 40 % of the references are older than five years. I believe than some very old references are not necessary.  

Author Response

Comments and Suggestions for Authors

Concerning the language, the manuscript is well written. Only, the title is not clear: what is the role of the word environmentally? Is something missing? Tables and figures (except the ordinate of figure 2) is well presented, too. Techniques applied are appropriate.  The idea is worth of investigation, unfortunately the results are not worth of publishing. Why? 

The targets pointed out in lines 98-104 were not achieved. Influence of tocopherol on compatibilization and plasticization, if there was no sample without tocopherol, is impossible to established. Other properties were worsened compared to control sample: processing properties, roughness, mass loss as well as the collagen production. Decrease of the glass transition temperature rather than the melting point is the measure of plasticization action. Let’s go step by step: 

We deleted the environmentally word from title. We hope that the title “Poly(3-hydroxybutyrate-co-3-hydroxyvalerate) (P(3HB-co-3HV))/bacterial cellulose (BC) biocomposites for potential use in biomedical applications” is better adequate with our research study.

Figure 2 represents the plotted of the mass loss in time. We added the title of OX axis.

  1. All samples contain 1 % of tocopherol, so the influence of tocopherol is not possible to determine. The only parameter that changes is the BC content. 

We performed the melt processability of neat PHBV, and its characterization in terms of AFM, DSC, FTIR and cell viability. Now, we hope that the addition of tocopherol can be observed for improving the melt processability and cell viability, and decreasing the roughness surface of PHBV and degree of crystallinity as compared with neat PHBV.

  1. From table 1 is obvious that addition of BC doesn’t improve the melt processing. 

Indeed, the BC filler doesn’t improve the melt processability of PHBV, but, in the presence of 1% tocopherol it did not decrease more than the neat PHBV, and the processing parameters are acceptable for conventional equipment of plastic materials.

  1. Also, from the table 2 is obvious that BC addition increase the surface roughness. 

Following text was added to Discussion section:

“The incorporation of BC filler led to an increased roughness surface of PHBV/BC biocomposited, but maintained in nanometer scale.”

Following text was added to Result section:

“The vitamin E contributes to a relatively smooth surface of PHBV film, assuring a better dispersion in polymeric matrix. These results are correlated with the processing parameters reported in Table 1.”

  1. Table 2 shows that with increase in BC content weigh loss increase, too. 

Following text were added to Discussion section:

“The addition of BC improved the mass loss recorded for PHBV/BC biocomposites, which is correlated with the surface roughness values of samples, attributed to the migration of BC from biocomposites. Similar behavior was reported for PHBV/highly crystalline cellulose nanoparticles (CNPs) composites, and explained by the creation of barrier layer due to the rigid hydrogen-bond network between components, able to protect the functional groups of PHBV [19].”

  1. DSC parameters (Table 3), except crystallinity, do not show significant changes. Taking in to account method accuracy and experimental errors, melting points are the same for all samples, as well as before and after immersion in PBS. Melting temperature decrease in not a measure of plasticization effect, the glass transition temperature is. But, as I wrote at the beginning, all of your samples contain tocopherol in the same amount. The melting point depression is the result of BC addition and it is normal, because BC can influence the copolymer crystallization. 

We agree that the melting temperature depression is a consequence of BC.

Following text were added to Results section:

“Data from Table 3 showed that both melting temperature peaks decreased for PHBV/BC1% and PHBC/BC2% biocomposites as compared with PHBV sample, suggesting the generation of immiscible biocomposites”.

“The degree of crystallinity of PHBV/Vitamin E sample decreases with respect to that of neat PHBV. This decrease is attributable to the vitamin E, which improves the flexibility of macromolecular chains but reduces the mobility of crystallizable segments from the PHBV chain. This behavior is in good agreement with the lower melt viscosity reported in Table 1.”

  1. All samples have almost the same cell compatibility, but the values of secreted collagen decrease with BC content. 

Indeed, the collagen synthesis decreased with the BC content, but this decrease is quite low, suggesting that the content of BC has no important effect on collagen synthesis. Nevertheless, the obtained results showed that the composite materials containing also BC stimulated cell proliferation and collagen production compared to the untreated cells.

  1. Discussion in the lines 421-422 is not the result of presented investigations: melt viscosity was increased; energy consumption was higher… 

“The adding of vitamin E enhanced the processing characteristics of PHBV natural polyester, while the addition of BC to polymer matrix did not lead to hard processing conditions.”

  1. 23 % of cited references are older than ten years and 40 % of the references are older than five years. I believe than some very old references are not necessary.  

We deleted as much as possible the old references, and introduces the most recent ones.

Round 2

Reviewer 3 Report

Dear authors,

thank you for the explanations and great effort you made for make the manuscript better.

And at last: in line 613 add the word parameters after melt processing.